# Asymmetric Influence of Dual-Task Interference on Anticipatory Postural Adjustments in One-Leg Stance

**DOI:** 10.3390/ijerph191811289

**Published:** 2022-09-08

**Authors:** Young Hoon Song, Si Ni Cho, Soo Mi Nam

**Affiliations:** Department of Physical Education, Seoul National University, Seoul 08826, Korea

**Keywords:** anticipatory postural adjustments, dual-task interference, one-leg stance, PCA

## Abstract

This study investigated the differences of anticipatory postural adjustments (APAs) in a one-leg stance (OLS) that appear according to lower-extremity dominance and dual-task interference. Thirteen young, healthy, male volunteers performed the OLS task under the following six conditions: (1) dominant leg (DL), single-task; (2) DL, dual-task, with a low level of cognitive load (DT1/2); (3) DL, dual-task, with a high level of cognitive load (DT + 1); (4) non-dominant leg (NDL), single-task; (5) NDL, DT1/2; and (6) NDL, DT + 1. In order to measure the subjects’ APA, we used the medial–lateral displacement of their centers of pressure and gravity from the force plate and the time-series data of joint angular motions, recorded using a 3D motion analysis system. In the NDL under the dual-task condition, the onset of APA was delayed and the amplitude declined, which resulted in an increase in the duration of the APA period. The number of components identified by principal component analysis differed according to the dominant foot, and the change caused by cognitive load was found only in the NDL. As the cognitive load increased, the variance of the principal component decreased. These findings show that dual-task interference asymmetrically influences APA according to limb dominance, which reorganizes the coordination strategy of joints’ angular motion.

## 1. Introduction

Human beings have evolved through the development of locomotion skills such as walking and running, based on the stability of vertical posture using both feet. However, the upright posture in humans is structurally unstable owing to the high location of the center of mass (COM), the small base of support (BOS), and the many joint actions formed between the BOS and COM [1]. To maintain equilibrium of the standing position, the COM of the body is located in the BOS and is required to align with the center of pressure (COP) [2].

The one-leg stance (OLS) used in the clinical assessment of balance is a simple and useful measurement method for evaluating postural stability, which is necessary to achieve daily activities such as walking and climbing [3,4]. The OLS task consists of two phases: (1) an initial dynamic balance phase followed by (2) static balance [5]. In the first phase, there is a transition movement from two- to one-leg standing. The goal of the second phase is to maintain body alignment with one leg. Most clinical OLS tests focus on the second static phase, although the initial dynamic balance phase can also be a decisive cue of postural stability [6].

In order to stand on one leg, anticipatory postural adjustments (APAs) must precede. APAs are a strategy used by the central nervous system that minimizes forthcoming postural perturbations, and occurs before the onset of voluntary movement [7,8]. If the anticipatory corrections are not made during switching from double-leg to single-leg standing, disequilibrium torque occurs in the direction of the lifting foot, and thus, standing on one foot cannot be maintained. While performing the OLS task, the APA prior to the one-leg stance controls a shift in the COP toward the lifting foot, which causes the center of gravity to accelerate toward the supporting foot. APAs are usually explained in terms of their amplitude and their duration on the basis of the COP [9,10].

APAs during single-leg standing have been studied extensively in healthy children [11], young adults [12,13], older adults [14], people with neurological diseases [15], and individuals with orthopedic diseases [16]. Since most studies of APAs related to the OLS task have used dominant or preferred feet, assuming the symmetry of both feet [17,18], the difference in lower-limb dominance remains unclear. Some studies have suggested asymmetrical APA between both feet caused by a pathological impairment [19,20], while others have reported that results for asymmetrical APA were also observed in healthy, young adults [21,22]. Thus, this suggests that APA according to lower-limb dominance in healthy adult merits further scrutiny.

Postural control in a daily or sports environment is usually performed with at least one other concurrent cognitive task (e.g., standing while talking). As a person’s attentional resources are limited, it is necessary to allocate attention for performing the multiple tasks at the same time (e.g., attention is divided between motor and cognitive tasks) [23]. Some empirical evidence shows that the effect of postural control under the dual-task condition decreases in comparison with single-task performance due to dual-task interference [24]. However, other studies have found that performing dual tasks during postural control improves balance [25]. It is suggested that dual tasks cause the dispersion of resources, resulting in automatic postural adjustments. Thus, the current empirical inconsistency in postural control while dual-tasking suggests that the relationship between postural control and cognitive processing deserves further research.

Therefore, taken together, the aim of this study is to investigate the differences in anticipatory postural adjustments in OLS that appear according to lower-extremity dominance and dual-task interference.

## 2. Methods

### 2.1. Participants

Thirteen young, healthy, male volunteers, without any history of neuromuscular disorder or balance pathology that might influence the OLS task, were recruited for this study. The participants had a mean age of 25.62 years (SD = 3.36), a mean height of 176.54 cm (SD = 5.91), and a mean mass of 74.60 kg (SD = 8.22). Among them, 11 participants were right-foot dominant and 2 were left-foot dominant; they were examined using the Revised Waterloo Footedness Questionnaire [26]. The experimental procedures were approved by the Seoul National University Institutional Review Board (IRB #1806/011-009, 11 June 2018), and all participants signed their written informed consent before the testing. 

### 2.2. Apparatus

A Qualisys Track Manager (Qualisys AB, Göteborg, Sweden) that records the motion of the retroreflective markers (super-spherical markers, Ø 14 mm, Qualisys, Sweden) attached over the anatomical landmarks of the subjects was used to derive three-dimensional kinematic data. Prior to the experiments, 14 markers were attached at anatomical joint landmarks on the head, shoulder, hip, knee, ankle, and toe, respectively (Figure 1). Eight infrared cameras (Opus 500, Qualisys, Sweden) dangling uniformly from the ceiling captured all of the markers on the participants at any given time. A single force platform (Bertec, Columbus, OH, USA) placed on the floor was used to investigate the displacement of the COP in anterior–posterior (COPap) and medial–lateral (COPml) directions. The 3D motion capture system and the force platform were synchronized with each other and sampled data at 100 Hz. All data were filtered by a low-pass fourth-order Butterworth filter with cut-off frequency of 10 Hz. 

### 2.3. Procedures

The digit span backward test, as a measure of working memory, was not only conducted to set the appropriate cognitive load for the participants, but was also offered as a concurrent task during the OLS task. The subjects were first asked to remember a series of numbers (e.g., 5, 2, 7) presented for 500 ms, and then, recall the digits in reverse serial order. The digit span backward test started with a sequence of 2 digits. If the individuals could respond correctly, the number of digits was gradually increased. When they answered incorrectly, the test was stopped. The maximum number of digits the participants could recall without error was regarded as the capacity of the working memory. Through this, we set up the level of cognitive load in the OLS task under the dual-task condition: low level (half the digits of their maximum digit number) and high level (added one number of their maximum digit number).

In the OLS task, the initial position was standing on the platform with the eyes open, with weight evenly distributed between both feet, and keeping the arms across the chest. There was a computer screen placed on a wall 3 m away from the platform at eye level, and subjects looked at it. After presenting all the digits on the screen, the subjects were instructed to stand one leg for more than 5 s. An auditory signal was directed at the participants when lifting one leg. The subjects recalled a series of digits in reverse order during the OLS task. 

The participants performed the OLS task under the following six conditions: (1) dominant leg, single-task (ST); (2) dominant leg, dual-task, with a low level of cognitive load (DT1/2); (3) dominant leg, dual-task, with a high level of cognitive load (DT + 1); (4) non-dominant leg, single-task; (5) non-dominant leg, DT1/2; and (6) non-dominant leg, DT + 1. Each condition was randomly assigned across all participants with the five trials blocked within a condition. A 1 min rest between trials and a 2 min rest between conditions was imposed to avoid the effects of fatigue.

### 2.4. Data Analysis

In order to measure the subjects’ APA, we used the ML displacement of their COP from the force plate as in previous studies [27,28,29] (Figure 2). The total displacements of COPml were calculated after subtracting the initial COP position, which was defined as the average COP position over the first 500 ms on the basis of the auditory signal. We identified the duration of APA as the time from which the COP began to displace toward the lifting leg (t0) until before a participant lifted a foot off the force plate (FO). The amplitudes of APA were defined as the peak ML displacement of the COP in the swinging leg before lifting. The velocity of the center of gravity (COG) along the ML direction was extracted from the ground reaction forces according to Newton’s second law.

The time-series data of the joint angular motions during the OLS task, determined using Principal component analysis (PCA), provided information on changes in the inter-joint coordination structure. First, the angular displacements of the trunk, pelvis, knee, and ankle in the frontal and sagittal planes were calculated. For the knee and ankle, the angles were calculated for both the L and supporting limb (S) sides. For each subject and each trial, a data matrix was formed from nine columns (nine joint motions) and normalized to 100 rows. To derive the principal components (PCs), 9 variables were demeaned to eliminate bias values of the larger amplitude of each variable. Then, the eigenvalue-eigenvector pairs of the covariance matrix were obtained. The percentage of total variability explained by each PC was also calculated. Of the nine components, any components that accounted for more than 11% of the variance were considered significant PCs. Finally, the mean loadings of all the subjects were examined and are represented as absolute values. A value larger than 0.4 indicates a significant contribution of each variable to the corresponding PC [30].

### 2.5. Statistical Analysis

A Box–Cox transformation was applied if the data were not normally distributed, as determined by the Shapiro–Wilk test. A two-way repeated-measures analysis of variance (ANOVA) as a within-subjects factor was used to investigate the differences in the lower-limb dominance (NDL or DL) and the level of cognitive load (ST, DT1/2 or DT + 1). The post hoc comparisons were conducted using a Holm–Bonferroni adjustment for multiple comparisons [31]. The Pearson correlation between the APA parameters was calculated. The alpha level was set at 0.05 to indicate significance.

## 3. Results

### 3.1. APA Parameters

Figure 3 represents the average APA parameters in the tests of limb dominance as a function of the cognitive load induced by the concurrent task. For APA onset (Figure 3A), the ANOVA revealed that there was a significant interaction between limb dominance and the task condition: *F*_2,24_ = 8.378, *p* = 0.013, and η^2^ = 0.318. The post hoc analysis revealed that the DT + 1 had a later onset than both the ST (*p* = 0.002) and DT1/2 (*p* = 0.013) in the NDL.

Regarding APA amplitude (Figure 3B), the ANOVA revealed that limb dominance (*F*_1,12_ = 7.940, *p* = 0.016, η^2^=0.398) and task condition (*F*_2,24_ = 3.970, *p* = 0.032, η^2^=0.249) had significant main effects on APA amplitude. However, there was no interaction effect between limb dominance and task condition. The post hoc analysis revealed that the DT + 1 had a smaller amplitude, compared to the ST, in the NDL (*p* = 0.008).

Regarding APA duration (Figure 3C), the ANOVA revealed that limb dominance (*F*_1,12_ = 5.228, *p* = 0.041, η^2^=0.303) and task condition (*F*_2,24_ = 5.836, *p* = 0.009, η^2^=0.327) had significant main effects on APA duration; however, there was no interaction effect between limb dominance and task condition. The post hoc analysis revealed that the DT1/2 had a longer duration than the ST in the NDL (*p* = 0.030).

For the COGvelo (Figure 3D), the ANOVA revealed that were no significant main effects or interaction effects.

### 3.2. Correlations between APA Parameters

Figure 4 presents the observed correlations between APA parameters. Significant correlations were observed between amplitude and duration (Figure 4A; R = −0.637, *p* < 0.001) and between amplitude and COGvelo (Figure 4B; R = 0.761, *p* < 0.001); however, there was no significant correlation between APA onset and duration (Figure 4C).

### 3.3. Variance Explained by PCs

Table 1 shows that the first two components (PC1 and PC2) accounted for more than 11% of the variance under each condition. Therefore, the rest of the PCA analysis focused on these two components. For the variance explained by PC1 (Figure 5A), the ANOVA revealed that there was a significant interaction effect between limb dominance and the task condition (*F*_2,24_ = 6.012, *p* = 0.004, η^2^=0.137). The post hoc analysis revealed that the DT + 1 had significantly larger variance than both the ST (*p* < 0.001) and DT1/2 (*p* = 0.002) in the NDL. For the variance explained by PC2 (Figure 5B), the ANOVA revealed that there was a significant interaction effect between limb dominance and the task (*F*_2,24_ = 5.528, *p* = 0.006, η^2^=0.127). The post hoc analysis revealed that the DT + 1 had significantly smaller variance than both STs in the NDL (*p* = 0.001 and DT1/2, *p* = 0.002).

### 3.4. Loadings of the PCs

The coordination pattern was examined based on the loading properties of the PCA. As mentioned above, the averaged loadings of the first two components were examined (Table 2). Under all conditions, the hip sagittal in PC1 and the L. knee sagittal in PC2 were strongly responsible for the movements. For DT1/2, the S. knee sagittal in PC2 was also found to be responsible for the movements. For DT + 1, the S. knee sagittal and S. ankle sagittal in PC1 contributed only in the NDL.

## 4. Discussion

In this study, we analyzed the effect of dual-task interference and foot dominance on the APA strategy during the OLS task. There are several key findings. First, we reveal a difference in APA depending on the lower-limb dominance. Second, dual-task interference appears only in the non-dominant leg. These main results are discussed in detail below.

The delayed onset of APA under the dual-task condition observed in this study is congruent with previous data from the literature on gait initiation [29] and choice reaction time compared to the simple reaction time condition [14]. This indicates that more time is required for cognitive processing to prepare APA while performing the OLS task with a concurrent task. Since postural control and cognitive activity share attentional resources [32], it is interpreted that the interference in resource allocation appeared under the dual-task condition.

In addition, the amplitude of APA is larger in the DL than in the NDL (which is preferred for supporting the body’s weight). Some studies have shown that postural asymmetry appeared in lower-limb movements in healthy subjects [12,22,33]. This may be due to differences in the preferred function of the lower extremities. In other words, lower-limb mobility (or manipulation) is generally considered to be associated with the DL, whereas the foot used to support (stabilize) motion is defined as the NDL [34,35]. Another important finding is that dual-task interference leads to a decrease in the amplitude of APA only seen under the NDL condition. A fear of falling while standing at the edge of a high platform has modified APA into a conservative strategy that causes a reduction in the magnitude of the posture and its potentially destabilizing effects on the direction of movement [36].

Unlike previous studies reporting no differences in APA duration between single-task and dual-task conditions [29,37], our result reveals that dual-task interference increased APA duration in healthy subjects. This discrepancy with the existing literature might be due to the insufficient cognitive load levels to induce dual-task interference, since there is a significant difference at the high level of cognitive load in the present study. 

There is a significant negative correlation between the amplitude and duration of APA as well as a positive correlation between APA amplitude and COGvelo. These results are consistent with the existing notion that a reduced volume is compensated by a longer period of APA, and that size is proportional to the velocity at which the weight is transferred to the supporting leg before lifting the leg [38].

In this study, the same principal components (PC) were extracted under all conditions, but as the cognitive load increased, the variance explained by PC 1 increased and, conversely, PC 2 decreased. Thus, it can be estimated that the use of the dynamic degree of freedom (DOF) is affected as the decreased attentional resource used for posture control because of the concurrent task. Similarly, the previous study confirmed the change in the coordination structure between joints and the reduction in the number of DOFs as the ground moved faster when the participants maintained their postural balance on a moving platform [39]. Therefore, it assumes that the complexity of the task causes the dispersion of attention resources, and the decrease in the resources allocated for postural control reduces the available dynamic DOF. The PCA loading shows that under all conditions, the hip and the knee joint of the lifting limb in the sagittal plane appear to be significant contributions to PC1 and PC2, respectively. The hip flexion in PC1 compensated for the upward acceleration that occurred when lifting the leg, whereas the knee flexion in PC2 generated a propulsive force, which is necessary to displace the COG through the ground reaction force. This is consistent with the role of APA in counteracting the forthcoming effect of perturbation in a feedforward fashion [8].

Additionally, under the condition of high cognitive load (DT + 1), asymmetry is found according to foot dominance. The knee and ankle joint of the supporting leg also contributed to PC1, which might explain the reduction in APA amplitude derived from the dual-task interference. This suggests that because of the flexion of the ankle and knee in the supporting leg, the lower ground reaction force generated from the lifting foot for shifting the weight was needed. Consequently, the dual-task interference changed the organizational properties of the coordination pattern of the APA. This may provide relevant interpretation for previous studies that reported a protective strategy during weight transfer in unstable subjects or environments [10,38,40].

The limitations of this study are the small sample size and the recruitment of only male subjects. As generalization is difficult due to the small number of samples, this should be supplemented in future studies. It has already been shown through previous studies that differences in APA are due to age, gender, and various diseases [11,12,13,14,15]. The reason we conducted the study only with healthy male adults is we thought that the third variables (age, gender, disease, etc.) should be controlled in order to check how the cognitive load, the subject of our study, affects APA.

## 5. Conclusions

This study reveals that dual-task interference has asymmetrical influences on the APA strategy of both legs during the OLS task. Collectively, these findings show that the NDL is more affected by the dual-task than the DL, which alter the joint coordination patterns. These results offer new insight into the nature and mechanisms of APA appearing under dual-task conditions.

## Figures and Tables

**Figure 1 ijerph-19-11289-f001:**
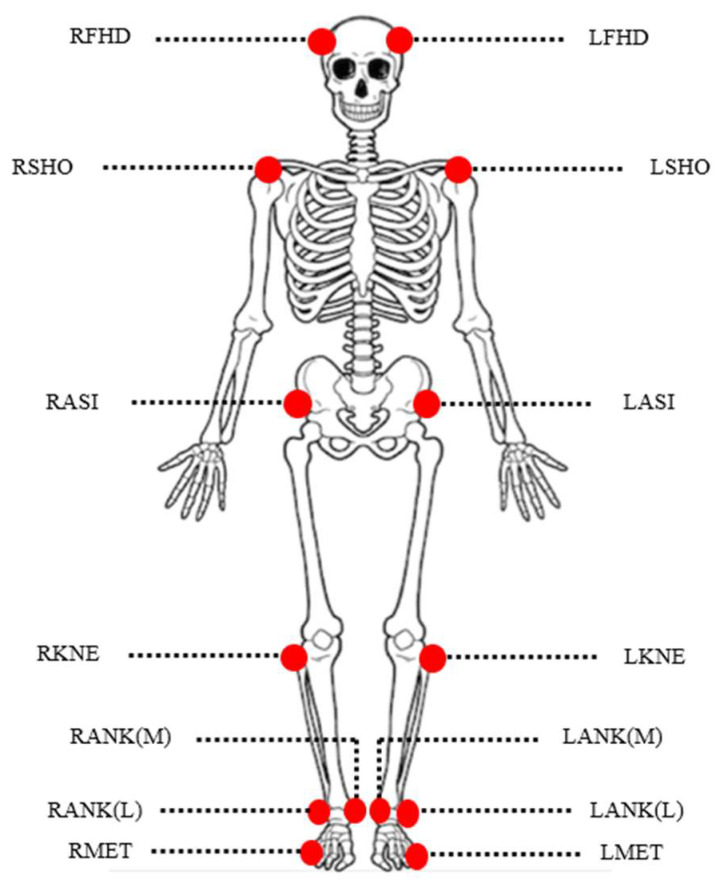
Reflective marker placement positions for the experiment. RFHD: located approximately over the right temple; LFHD: located approximately over the left temple; RSHO: placed on the right acromio-clavicular joint; LSHO: placed on the left acromio-clavicular joint; RASI: right anterior superior iliac spine; LASI: left anterior superior iliac spine; RKNE: placed on the lateral epicondyle of the right knee; LKNE: placed on the lateral epicondyle of the left knee; RANK(M): right medial malleolus; LANK(M): left medial malleolus; RANK(L): right lateral malleolus; LANK(L): left lateral malleolus; RMET: right 5th metatarsal head; LMET: left 5th metatarsal head.

**Figure 2 ijerph-19-11289-f002:**
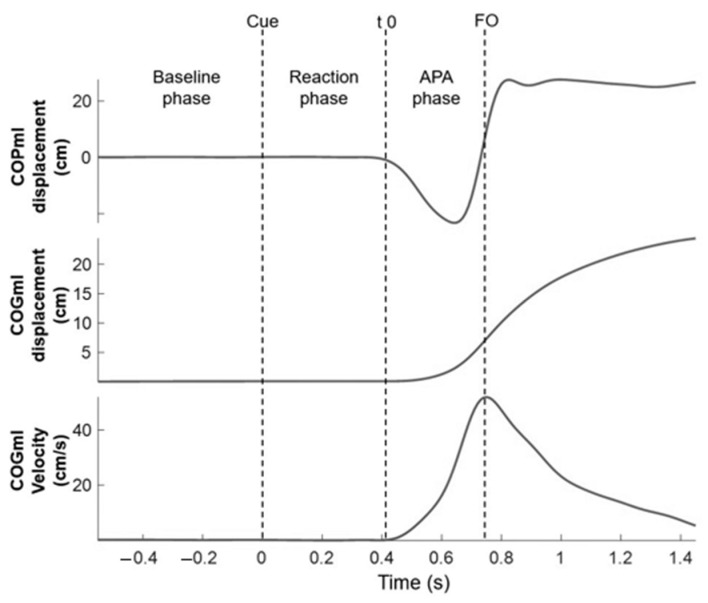
Representative trial showing the displacement of COP, and displacement and velocity of the center of gravity (COG) in the mediolateral direction. The events and phases are marked as follows. Cue: presentation of the auditory cue; t0: onset variation in the medial–lateral COP (COPml) from the baseline; FO: lifting foot off the ground (FO); baseline phase: 500 ms period prior to cue; reaction phase: time from cue to t0; and APA phase: time from t0 to FO.

**Figure 3 ijerph-19-11289-f003:**
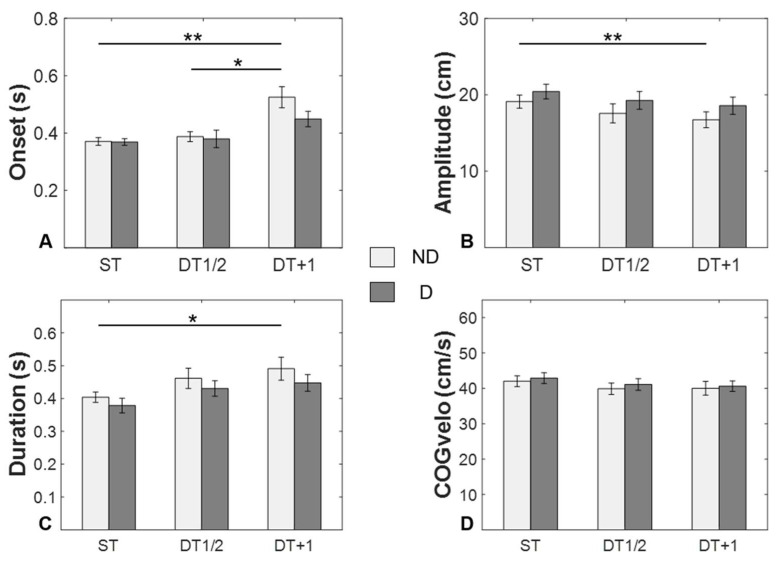
Comparison of measurement parameters between dominant (D) and non-dominant (ND) limbs and under different task conditions (single-task (ST), dual-task 1/2 (DT1/2), and dual task +1 (DT + 1)). The measurement parameters consisted of APA onset (**A**), APA amplitude (**B**), APA duration (**C**), and COG_velo_ (**D**). Values are mean ± standard error. There were significant post hoc comparisons (Holm–Bonferroni adjusted for multiple comparisons): * *p* < 0.05; ** *p* < 0.01.

**Figure 4 ijerph-19-11289-f004:**
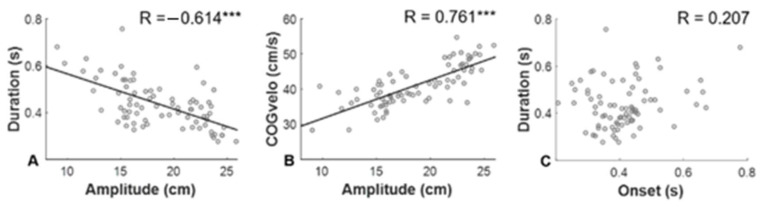
The relationship between measurement parameters. A circle represents the mean value of each condition for a participant. The correlation coefficient (R) and level of significance are indicated (*** *p* < 0.001). (**A**) APA amplitude negatively correlated with APA duration (R = −0.637, *p* < 0.001). (**B**) APA amplitude significantly correlated with COG_velo_ (R = 0.761, *p* < 0.001). (**C**) APA onset was not correlated with APA duration.

**Figure 5 ijerph-19-11289-f005:**
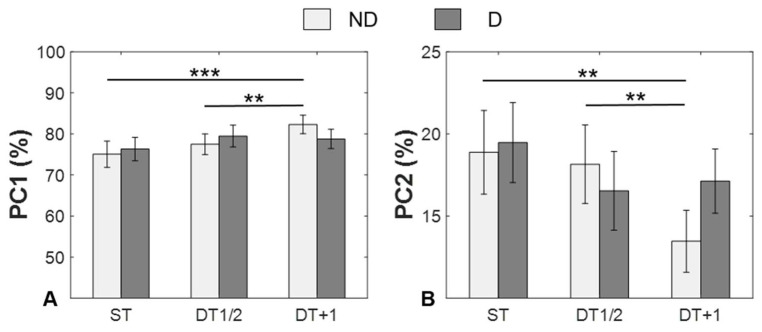
Comparison of variance explained by the first three principal components between ND and D limbs under different task conditions (ST, DT1/2, and DT + 1). Values are mean ± standard error. Significant post hoc comparisons (Holm–Bonferroni adjusted for multiple comparisons): ** *p* < 0.01. *** *p* < 0.001. There was a significant interaction effect between limb dominance and task condition in PC1 (**A**) and PC2 (**B**).

**Table 1 ijerph-19-11289-t001:** Mean percentage of variance explained by first two principal components.

Limb	Task	Variance (%)	PC1	PC2
ND	ST	Explained	75.06	18.89
		Accumulated	75.06	93.95
	DT1/2	Explained	76.32	19.47
		Accumulated	76.32	95.79
	DT + 1	Explained	77.48	18.15
		Accumulated	77.48	95.63
D	ST	Explained	79.47	16.54
		Accumulated	79.47	96.01
	DT1/2	Explained	82.29	13.47
		Accumulated	82.29	95.76
	DT + 1	Explained	78.76	17.13
		Accumulated	78.76	95.89

PC: principal components, ST: single-task, DT1/2: dual-task1/2, DT + 1: dual-task + 1, ND: non-dominant, and D: dominant.

**Table 2 ijerph-19-11289-t002:** Mean loadings presented in absolute values of the first two principal components.

	ST		DT 1/2		DT + 1	
	ND	D	ND	D	ND	D
Hip frontal						
PC1	0.26	0.22	0.25	0.17	0.22	0.20
PC2	0.14	0.17	0.15	0.17	0.15	0.17
Hip sagittal						
PC1	**0.42**	**0.42**	**0.41**	**0.46**	**0.49**	**0.46**
PC2	0.31	0.26	0.32	0.25	0.29	0.25
Pelvis frontal						
PC1	0.25	0.23	0.25	0.22	0.23	0.26
PC2	0.15	0.16	0.16	0.15	0.13	0.14
L. Knee sagittal						
PC1	0.34	0.37	0.36	0.33	0.32	0.33
PC2	**0.65**	**0.58**	**0.54**	**0.55**	**0.60**	**0.61**
S. Knee sagittal						
PC1	0.34	0.37	0.38	0.36	0.40	0.37
PC2	0.31	0.37	**0.44**	**0.45**	**0.40**	**0.44**
L. Ankle frontal						
PC1	0.10	0.07	0.09	0.06	0.08	0.07
PC2	0.12	0.15	0.15	0.13	0.12	0.12
S. Ankle frontal						
PC1	0.09	0.11	0.11	0.11	0.11	0.11
PC2	0.07	0.06	0.06	0.09	0.06	0.08
L. Ankle sagittal						
PC1	0.32	0.31	0.28	0.29	0.18	0.24
PC2	0.28	0.27	0.29	0.22	0.28	0.28
S. Ankle sagittal						
PC1	0.36	0.30	0.38	0.32	**0.40**	0.36
PC2	0.19	0.25	0.19	0.22	0.12	0.14

PC: principal components, ST: single-task, DT1/2: dual-task 1/2, DT + 1: dual-task + 1, ND: non-dominant, D: dominant, L: lifting-limb side, S: supporting-limb side, frontal: motion in the frontal plane, and sagittal: motion in the sagittal plane. Bold font indicates loadings over 0.4.

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
