# Peer review of "Asymmetric Influence of Dual-Task Interference on Anticipatory Postural Adjustments in One-Leg Stance"

_ijerph, 2022, doi:10.3390/ijerph191811289_

Round 1
Reviewer 1 Report
Interesting research which investigates the relationship between postural control and cognitive load.
Point 1. It was mentioned in the methodology section, that the Qualisys system was used to collect kinematic data but there is no information on which marker model was used (IOR or CAST) and what camera model. Could you add it to the methodology section?
Point 2. Have you done power analysis (statistical power) to determine if the sample size was enough as Bonferroni correction also increases type 2 error and you have small sample size?
Point 3. Could you more explain data filtration? All data it means kinematics data was filtered with 5 Hz cut-off frequency and force data? Why would you choose 5Hz? With the modern Qualisys system, 10 Hz is enough and Force data is usually filtered with a 15 or 25 cut-off frequency.
Author Response
First of all, I would like to express my sincere thanks to the reviewer for their valuable comments on this paper. All authors agreed with the revisions presented and faithfully revised them as follows in accordance with the points pointed out.
Point 1. It was mentioned in the methodology section, that the Qualisys system was used to collect kinematic data but there is no information on which marker model was used (IOR or CAST) and what camera model. Could you add it to the methodology section?
- It has been added according to the contents you pointed out.
Point 2. Have you done power analysis (statistical power) to determine if the sample size was enough as Bonferroni correction also increases type 2 error and you have small sample size?
- We consider the small number of subjects to be a limitation of this study (13 subjects, 5 trials). However, I would like to note that a previous study [2] that was similar to my research topic and used the same statistical technique also yielded meaningful results (8 subjects, 5 trials).
Point 3. Could you more explain data filtration? All data it means kinematics data was filtered with 5 Hz cut-off frequency and force data? Why would you choose 5Hz? With the modern Qualisys system, 10 Hz is enough and Force data is usually filtered with a 15 or 25 cut-off frequency.
- We agree with your point. In the data processing, it was recommended to filter at 10Hz considering joint movement in units of 0.1 second, so we modified it from 5Hz to 10Hz. However, in this paper, it is written as the original plan, but was actually filtered at 10Hz. As a result of analyzing both kinematics and force data through Fast Fourier Transformation, values above 10 Hz were insignificant enough to not affect the results, so both data were unified to 10 Hz. Therefore, it has been modified.
Reviewer 2 Report
Thank you for submitting this paper to IJERPH. The manuscript under consideration: "Asymmetric Influence of Dual-Task Interference on Anticipatory Postural Adjustments in One-Leg Stance" is an interesting article on an important topic in IJERPH. However, there are a few major concerns.
1. OLS is largely related to muscle strength as well as balance function. Also, depending on the age of the participant, there may not be a significant difference between left and right OLS times.
This suggests that the difference is first of all due to differences in muscle strength, age, gender, and static balance function rather than left-right differences due to APA. The background of the present authors' study is too poorly explained.
2. Why did you target 20-somethings in researching this study? Also, the sample size is too small and I don't see the significance of investigating the difference in OLS's and APA's from this age to the left and right.
3. The site of application of the retroreflective marker is unclear. Please explain in detail.
4. Which equipment was used for the three-dimensional motion analyzer?
5. I wonder, why did you validate 6 conditions, if you are validating left-right differences in APA, wouldn't basic OLS suffice? Also, is the assignment appropriate for 20-somethings?
6. Inclusion and exclusion criteria are not clear.
7. The "Shapiro-Wilk test" should be used instead of the "Kolmogorov-Smirnov test" to check the normality distribution.
8. OLS explains that it is more than 5 seconds, but there are many assignments that are exactly 120 seconds, and the definition of more than 5 seconds is an inaccurate OLS assignment teaching.
9. I don't see how this is different from previous APA studies, just that the issue is now OLS. What is the novelty of this study? As I mentioned at the beginning, I don't think the left-right difference in OLS can be explained by APA alone.
Author Response
First of all, I would like to express my sincere thanks to the reviewer for their valuable comments on this paper. All authors agreed with the revisions presented and faithfully revised them as follows in accordance with the points pointed out.
1.OLS is largely related to muscle strength as well as balance function. Also, depending on the age of the participant, there may not be a significant difference between left and right OLS times.
This suggests that the difference is first of all due to differences in muscle strength, age, gender, and static balance function rather than left-right differences due to APA. The background of the present authors' study is too poorly explained.
- This study is about effect of cognitive load on left and right APA using kinematic and force data during OLS, not about how much APA affects OLS. The relevant information in the introduction is as follows.
Postural control in daily or sport environment is usually performed with at least one other concurrent cognitive task (e.g., standing while talking). As a person’s attentional resources are limited, it is required to allocate attention allocation for performing the multiple tasks at the same time (e.g., attention is divided between the motor and cognitive tasks). Some empirical evidences show that the effect of postural control under the dual-task condition decreases in comparison with single-task performance due to dual-task interference [16]. However, other studies found that performing dual tasks during postural control improves keeping balance. It is suggested that dual tasks cause dispersion of resources, resulting in the automatic postural adjustments [17]. Thus, the current empirical inconsistency in postural control while dual-tasking suggests that the relation between postural control and cognitive processing is worth further research.
2.Why did you target 20-somethings in researching this study? Also, the sample size is too small and I don't see the significance of investigating the difference in OLS's and APA's from this age to the left and right.
- It has already been shown through previous studies that difference in APA is due to age, gender, and various diseases. Therefore, the reason we conducted the study only with healthy male adults is because we thought that the third variables (age, gender, disease, etc.) should be controlled in order to check how the cognitive load, the subject of our study, affects the APA strategy on left and right.
In addition, we consider the small number of subjects to be a limitation of this study (13 subjects, 5 trials). However, I would like to note that a previous study [2] that was similar to my research topic and used the same statistical technique also yielded meaningful results (8 subjects, 5 trials).
3.The site of application of the retroreflective marker is unclear. Please explain in detail.
- It has been modified according to the contents you pointed out.
4.Which equipment was used for the three-dimensional motion analyzer?
- 8 infrared motion analysis cameras (Oqus 500, Qualisys, Sweden) were utilized to record 14 anatomical markers. The relevant information has been added.
5.I wonder, why did you validate 6 conditions, if you are validating left-right differences in APA, wouldn't basic OLS suffice? Also, is the assignment appropriate for 20-somethings?
- In order to examine the left-right difference on the APA strategy when a cognitive load was given along with the OLS task, our study was conducted under six conditions according to the dominance of the foot and the amount of cognitive load.
The reason we conducted the study only with healthy male adults is because we thought that the third variables (age, gender, disease, etc.) should be controlled in order to check how the cognitive load, the subject of our study, affects the APA strategy on left and right.
6.Inclusion and exclusion criteria are not clear.
- As in answer to the previous question, our study was conducted under six conditions according to the dominance of the foot and the amount of cognitive load. The specific conditions are as follows: 1) dominant leg, single-task (ST); 2) dominant leg, dual-task with low level of cognitive load (DT1/2); 3) dominant leg, dual-task, high level of cognitive load (DT+1); 4) non-dominant leg, single-task; 5) non-dominant leg, DT1/2; 6) non-dominant leg, DT+1.
7.The "Shapiro-Wilk test" should be used instead of the "Kolmogorov-Smirnov test" to check the normality distribution.
- As you mentioned, we checked that of the two, the Shapio-Wilk test is the more common. However, rather than focusing on the difference between the two test methods, please consider that the Kolmogorov-Smirnov test was used in the referenced previous study to confirm the normality distribution [3, 10].
8.OLS explains that it is more than 5 seconds, but there are many assignments that are exactly 120 seconds, and the definition of more than 5 seconds is an inaccurate OLS assignment teaching.
- Our research is about APA for standing on one foot, not postural control while standing on one foot. All data of this study are kinematic and force data during APA (about 0.5 seconds). The reason why we were asked to stand for 5 seconds in this study was that the APA occurring when participants accidentally dropped their leg while lifting one foot was not in line with the intention of this study.
9.I don't see how this is different from previous APA studies, just that the issue is now OLS. What is the novelty of this study? As I mentioned at the beginning, I don't think the left-right difference in OLS can be explained by APA alone.
- As you said, the APA can be affected for a variety of reasons. Therefore, in our study, only healthy male subjects were selected to control for this reason. The novelty of this study is that, first, it tries to explain the difference in APA with OLS with kinematic data, and second, it reveals that cognitive load affects the left and right APA differently.
Reviewer 3 Report
Dear I realize that authors have many journals to consider when they want to publish their work, so I appreciate your interest in Int. J. Environ. Res. Public Healths; I am very sorry not to be able to write in a more positive way. It is evident that you have put a great deal of effort into this project and I want to praise your efforts, Fortunately, the actual contribution from your study is clear and, the manuscript as currently written suggests that it might be suitable for sharing information about this topic, but the data that you reported, needs few edits. I should like to thank you for give me an opportunity to consider this work for publication. It may be that the you would like to consider resubmitting it, in which case I hope that the comments from my review may help you to revise it before resubmitting it. These comments are given below. Best Regards - Introduction section: references are missing in many sentences; - methods:
you should provide additional information on how the participants was recruited;
enter the number and date of authorization of the Seoul National University Institutional Review Board;
enter all the details of the infrared cameras and markers [type, model, brand, manufacturer, etc.]
- in discussion section:
Discussions should be reviewed in light of the overall improvement of the paper;
Redundant sentences and prewritten information should be avoided;
Focus on take-home messages and how that information impacts in the clinical practice;
insert a limitations section in the discussion, in which all the limits of the study are described in detail, such as the small sample number, the male-only sex of the subjects, etc.
- Figures: inserting a photo of the recruited subjects with markers while they perform the tasks would help to greatly improve the quality of the paper
Author Response
First of all, I would like to express my sincere thanks to the reviewer for their valuable comments on this paper. All authors agreed with the revisions presented and faithfully revised them as follows in accordance with the points pointed out.
- Introduction section: references are missing in many sentences;
- It has been added according to the contents you pointed out.
- methods: you should provide additional information on how the participants was recruited; enter the number and date of authorization of the Seoul National University Institutional Review Board; enter all the details of the infrared cameras and markers [type, model, brand, manufacturer, etc.]
- It has been added according to the contents you pointed out.
- in discussion section: Discussions should be reviewed in light of the overall improvement of the paper; Redundant sentences and prewritten information should be avoided; Focus on take-home messages and how that information impacts in the clinical practice; insert a limitations section in the discussion, in which all the limits of the study are described in detail, such as the small sample number, the male-only sex of the subjects, etc.
- According to the content you pointed out, duplicate sentences have been deleted and restrictions have been added.
- Figures: inserting a photo of the recruited subjects with markers while they perform the tasks would help to greatly improve the quality of the paper
- We have inserted a figure of the placement of the reflective markers according to the contents you pointed out (Fig. 1).
Round 2
Reviewer 2 Report
The inconsistency throughout this manuscript, especially in sample size, appropriate data collection and statistical analysis, renders the results and discussion of this study less meaningful.
1. As I think I have pointed out before, the inclusion and exclusion criteria are opaque. The authors' response does not provide a clear answer to the exclusion and inclusion criteria.
2. The sample size setting is unclear.
3. Why did authors select male only? From the male and female, it is possible to obtain measurements of APA on difference in sex. Authors should more clearly describe the purpose of selecting male only.
4. As I have pointed out before, the normal distribution is calculated incorrectly. The Shapiro-Wilk test is the appropriate method for this study, regardless of previous studies. The authors should be more careful in their statistics with more precise procedures.
5. I would recommend authors to conduct a larger study before any conclusion can be made for these experiments. At this point, the claims made in this manuscript is not supported by the data and is hypothetical in nature.
Author Response
First of all, I would like to express my sincere thanks to the reviewer for their valuable comments on this paper. All authors agreed with the revisions presented and faithfully revised them as follows in accordance with the points pointed out.
- As I think I have pointed out before, the inclusion and exclusion criteria are opaque. The authors' response does not provide a clear answer to the exclusion and inclusion criteria.
- We set conditions based on the following criteria. First, according to the foot dominance, it was divided the condition into dominant and non-dominant leg condition. Second, it is divided into single task and double task conditions in accordance with the number of tasks. Third, according to the individual's cognitive load measured through the digit span backward test, it was divided into low level and high level conditions. The relevant information in the methods is as follows.
Digit span backward test as a measure of working memory not only was conducted to set the appropriate cognitive load for the participants, but also offered as a concurrent task during the OLS task. The subjects were first asked to remember a series of numbers (e.g., 5, 2, 7) presented for 500ms and then recall the digits in reverse serial order. The digit span backward starts with a sequence of 2 digits. If the individuals could respond correctly, the number of digits was gradually increased. When they incorrectly reacted, the test was stopped. The maximum number of digits the participants can recall without error was regarded as the capacity of the working memory. Through this, we set up the level of cognitive load in the OLS with dual-task condition: low level (half digits of their maximum digit number) and high level (added one number of their maximum digit number).
In the OLS task, the initial position was standing on the platform with eyes open, weight evenly distributed between both feet, and keeping arms across their chest. There was a computer screen placed on a wall 3m away from the platform at eye level, and they looked at it. After presenting all digits on the screen, the subjects were instructed to standonelegformorethan 5 seconds.An auditory signal directed to the participants when lifting one leg. The subjects recalleda series ofdigitsinreverseorder while the OLS task.
The participants were performed the OLS task under the six conditions: 1) dominant leg, single-task (ST); 2) dominant leg, dual-task with low level of cognitive load (DT1/2); 3) dominant leg, dual-task, high level of cognitive load (DT+1); 4) non-dominant leg, single-task; 5) non-dominant leg, DT1/2; 6) non-dominant leg, DT+1. Each condition was randomly assigned across all participants with the five trials blocked within a condition. 1-min rest between trials and 2-min rest between conditions was imposed to avoid the effects of fatigue.
- The sample size setting is unclear
- As mentioned in the discussion section, we consider the small number of subjects to be a limitation of this study. Experimental research that has been verified with a small number of samples can be advanced by increasing the number of samples or changing conditions. Although this study was conducted with a small sample size, future studies will supplement this according to the reviewers' opinions.
- Why did authors select male only? From the male and female, it is possible to obtain measurements of APA on difference in sex. Authors should more clearly describe the purpose of selecting male only.
- It has already been shown through previous studies that difference in APA is due to age, gender and various diseases [11-15]. In other words, gender could be a potential covariate that could influence differences in APA. Therefore, to control for the influence of covariates, we studied only healthy male adults. As a result, it was possible to clearly confirm the difference in APA according to cognitive load. However, in future research, we will conduct the study considering the gender based on the results of this study by reflecting the opinions of the reviewer. The relevant information in the discussion is as follows.
The limitations of this study are the small sample size and the recruitment of only male subjects. As generalization is difficult due to the small number of samples, this should be supplemented in future study. It has already been shown through previous studies that difference in APA is due to age, gender, and various diseases [11-15]. The reason we conducted the study only with healthy male adults is because we thought that the third variables (age, gender, disease, etc.) should be controlled in order to check how the cognitive load, the subject of our study, affects APA.
- As I have pointed out before, the normal distribution is calculated incorrectly. The Shapiro-Wilk test is the appropriate method for this study, regardless of previous studies. The authors should be more careful in their statistics with more precise procedures.
- As you mentioned, we found that the Shapiro-Wilks test is valid for our paper sample size. Therefore, we performed the normality test again with the Shapiro-Wilks test. As a result, it was derived that the APA onset variable did not satisfy normality. However, since this result shows the same result as the previously performed normality test, it was modified that the normality test was performed with the Shapiro-Wilks test. The table below is the Shapiro-Wilks test result
|
Variables |
W |
p |
|
Onset |
0.94553 |
0.002256 |
|
Amplitude |
0.97971 |
0.2484 |
|
Duration |
0.97174 |
0.08014 |
|
COGvelo |
0.96849 |
0.05023 |
|
PC1 |
0.98093 |
0.2933 |
|
PC2 |
0.9714 |
0.07633 |
- I would recommend authors to conduct a larger study before any conclusion can be made for these experiments. At this point, the claims made in this manuscript is not supported by the data and is hypothetical in nature.
- Thank you for your valuable comments for the further development of this study. Due to the specificity of experimental research, there are cases in which a step-by-step research method is used to increase the number of samples from a small number of samples or to develop from a simple condition to a more complex condition. Please consider the possibility of future research based on the results of this study. Although there are limitations in the small sample size and only male subjects, please consider the following novelties. First, it tries to explain the difference in APA with OLS with kinematic data, and second, it reveals that cognitive load affects the left and right APA differently.